# Visual Prompting for Adversarial Robustness

**Aochuan Chen**[*]
Michigan State University
chenaoch@msu.edu

**Peter Lorenz**[*]
Fraunhofer ITWM[†]
peter.lorenz@itwm.fhg.de

**Yuguang Yao**
Michigan State University
yaoyugua@msu.edu

**Pin-Yu Chen**
IBM Research
pin-yu.chen@ibm.com

**Sijia Liu**
Michigan State University
liusiji5@msu.edu

## Abstract

In this work, we leverage visual prompting (VP) to improve adversarial robustness of a fixed, pre-trained model at testing time. Compared to conventional adversarial defenses, VP allows us to design universal (*i.e.*, data-agnostic) input prompting templates, which have plug-and-play capabilities at testing time to achieve desired model performance without introducing much computation overhead. Although VP has been successfully applied to improving model generalization, it remains elusive whether and how it can be used to defend against adversarial attacks. We investigate this problem and show that the vanilla VP approach is *not* effective in adversarial defense since a universal input prompt lacks the capacity for robust learning against sample-specific adversarial perturbations. To circumvent it, we propose a new VP method, termed Class-wise Adversarial Visual Prompting (C-AVP), to generate class-wise visual prompts so as to not only leverage the strengths of ensemble prompts but also optimize their interrelations to improve model robustness. Our experiments show that C-AVP outperforms the conventional VP method, with 2.1× standard accuracy gain and 2× robust accuracy gain. Compared to classical test-time defenses, C-AVP also yields a 42× inference time speedup. Code is available at github.

## 1 Introduction

Current machine learning (ML) models, *e.g.*, vision models in particular, can easily be manipulated (by an adversary) to output drastically different classifications and can be done so in a controlled and directed way. This process is known as *adversarial attack* and is considered as one of the major hurdles in using ML models in high-stakes applications [Goodfellow et al., 2014, Carlini and Wagner, 2017]. Thereby, model robustification against adversarial attacks is now a major focus of research. Yet, a large volume of existing works focused on advancing training recipes and/or model architectures to gain robustness. For example, adversarial training (AT) [Madry et al., 2017], one of the most effective defense methods, adopted min-max optimization to minimize the worst-case (maximum) training loss induced by adversarial attacks. Extended from AT, various empirical and certified defense methods were proposed in various learning paradigms, ranging from supervised learning to semi-supervised learning, and further to unsupervised learning [Zhang et al., 2019b, Shafahi et al., 2019, Zhang et al., 2019a, Carmon et al., 2019, Wong and Kolter, 2017, Raghunathan et al., 2018, Xie et al., 2019, Chen et al., 2020, Fan et al., 2021].

Although the design for robust training has made tremendous success in improving model robustness [Athalye et al., 2018, Croce and Hein, 2020], it typically takes an intensive computation cost with

---

[*]Equal Contribution. [†] and Fraunhofer Center of Machine Learning

2022 Trustworthy and Socially Responsible Machine Learning (TSRML 2022) co-located with NeurIPS 2022.

poor defense scalability to a fixed, pre-trained ML model. Towards circumventing this difficulty, the problem of test-time defense arises; see the seminal work in [Croce et al., 2022]. Test-time defense alters either a test-time input example or a small portion of the pre-trained model for adversarial defense. Examples include input (anti-adversarial) purification [Yoon et al., 2021, Mao et al., 2021, Alfarra et al., 2022] and model refinement by augmenting the pre-trained model with auxiliary components [Salman et al., 2020, Gong et al., 2022, Kang et al., 2021]. However, these defense techniques inevitably raise the inference time and hamper the test-time efficiency [Croce et al., 2022]. Inspired by that, our work will advance the test-time defense technology by leveraging the idea of *visual prompting* (**VP**) [Bahng et al., 2022], also known as model reprogramming [Chen, 2022, Elsayed et al., 2018, Tsai et al., 2020, Zhang et al., 2022].

Generally speaking, VP [Bahng et al., 2022] creates a *universal* (*i.e.*, *data-agnostic*) input prompting template (in terms of input perturbations) in order to improve the generalization ability of a pre-trained model when incorporating such a visual prompt into test-time examples. It enjoys the same idea as model reprogramming [Chen, 2022, Elsayed et al., 2018, Tsai et al., 2020, Zhang et al., 2022] or unadversarial example [Salman et al., 2021], which optimizes the universal perturbation pattern to maneuver (*i.e.*, reprogram) the functionality of a pre-trained model towards the desired criterion, *e.g.*, cross-domain transfer learning [Tsai et al., 2020], out-of-distribution generalization [Salman et al., 2021], and fairness [Zhang et al., 2022]. However, it remains elusive whether or not VP could be designed as an effective solution to adversarial defense. We will investigate this problem, which we call *adversarial visual prompting* (**AVP**), in this work. Compared to conventional test-time defense methods, AVP will significantly reduce the inference time overhead since visual prompts can be designed offline over training data and have the plug-and-play capability applied to any testing data. We summarize our **contributions** below.

❶ We formulate and investigate the problem of AVP for the first time. We empirically show that the conventional data-agnostic VP design is incapable of gaining adversarial robustness.

❷ We propose a new VP method, termed class-wise AVP (**C-AVP**), which produces multiple, class-wise visual prompts with explicit optimization on their couplings to gain adversarial robustness.

❸ We provide insightful experiments to demonstrate the pros and cons of VP in adversarial defense.

## 1.1   Related work

**Visual prompting.**   Originated from the idea of in-context learning or prompting in natural language processing (NLP) [Brown et al., 2020, Li and Liang, 2021, Radford et al., 2021], VP was first proposed in [Bahng et al., 2022] for vision models. Before formalizing VP in [Bahng et al., 2022], the underlying prompting technique has also been devised in computer vision (CV) with different naming. For example, VP is closely related to *adversarial reprogramming* or *model reprogramming* [Elsayed et al., 2018, Chen, 2022, Tsai et al., 2020, Neekhara et al., 2022, Yang et al., 2021, Zheng et al., 2021], which focused on altering the functionality of a fixed, pre-trained model across domains by augmenting test-time examples with an additional (universal) input perturbation pattern. *Unadversarial learning* also enjoys the similar idea to VP. In [Salman et al., 2021], unadversarial examples that perturb original ones using 'prompting' templates were introduced to improve out-of-distribution generalization. Yet, the problem of VP for adversarial defense is under-explored.

**Adversarial defense.**   The lack of adversarial robustness is a weakness of ML models. Adversarial defense, such as adversarial detection [Grosse et al., 2017, Yang et al., 2019, Metzen et al., 2017, Meng and Chen, 2017, Wójcik et al., 2020, Gong et al., 2022] and robust training [Wong and Kolter, 2017, Zhang et al., 2019b, Salman et al., 2020, Chen et al., 2020, Boopathy et al., 2020, Fan et al., 2021], is a current research focus. In particular, adversarial training (AT) [Madry et al., 2017] is the most widely-used defense strategy and has inspired many recent advances in adversarial defense [Athalye et al., 2018, Ye et al., 2019, Croce and Hein, 2020, Mohapatra et al., 2020, Kang et al., 2021, Wang et al., 2021]. However, these AT-type defenses (with the goal of robustness-enhanced model training) are computationally intensive due to min-max optimization over model parameters. To reduce the computation overhead of robust training, the problem of test-time defense arises [Croce et al., 2022], which aims to robustify a given model via lightweight unadversarial input perturbations (*a.k.a* input purification) [Shi et al., 2021, Yoon et al., 2021] or minor modifications to the fixed model [Chen et al., 2021]. In different kinds of test-time defenses, the most relevant work to ours is anti-adversarial perturbation [Alfarra et al., 2022].

## 2 Problem Statement

In this section, we will begin by providing a brief background on VP, and then introduce the problem of our interest–*adversarial visual prompting* (**AVP**)–which aims at generating visual prompts to improve adversarial robustness of a pre-trained, fixed model. Through a warm-up example, we will empirically show that the conventional design of VP is difficult to apply to the paradigm of AVP.

**Visual prompting.** We describe the problem setup of VP following [Bahng et al., 2022, Elsayed et al., 2018, Tsai et al., 2020, Zhang et al., 2022]. Specifically, let $\mathcal{D}_{\mathrm{tr}}$ denote a training set for supervised learning, where $(\mathbf{x}, y) \in \mathcal{D}_{\mathrm{tr}}$ signifies a training sample with feature $\mathbf{x}$ and label $y$. And let $\boldsymbol{\delta}$ be a visual prompt to be designed. The prompted input is then given by $\mathbf{x} + \boldsymbol{\delta}$ with respect to (w.r.t.) $\mathbf{x}$. Different from the problem of adversarial attack generation that optimizes $\boldsymbol{\delta}$ for erroneous prediction, VP drives $\boldsymbol{\delta}$ to minimize the performance loss $\ell$ of a pre-trained model $\boldsymbol{\theta}$. This leads to

$$\begin{array}{ll} \underset{\boldsymbol{\delta}}{\text{minimize}} & \mathbb{E}_{(\mathbf{x}, y) \in \mathcal{D}_{\mathrm{tr}}}[\ell(\mathbf{x} + \boldsymbol{\delta}; y, \boldsymbol{\theta})] \\ \text{subject to} & \boldsymbol{\delta} \in \mathcal{C}, \end{array} \tag{1}$$

where $\ell$ denotes a certain performance loss (*e.g.*, prediction error [Bahng et al., 2022]) given the prior knowledge of training data $(\mathbf{x}, y)$ and base model $\boldsymbol{\theta}$, and $\mathcal{C}$ is a perturbation constraint. Following [Elsayed et al., 2018, Tsai et al., 2020, Bahng et al., 2022], $\mathcal{C}$ restricts $\boldsymbol{\delta}$ to be located in an image's boundary region and requests the perturbation magnitude within a normalized input space, *i.e.*, $\mathbf{x} + \boldsymbol{\delta} \in [0, 1]$ for any $\mathbf{x}$. Projected gradient descent (PGD) [Madry et al., 2017, Salman et al., 2021] can then be applied to solving problem (1). At inference time, the designed $\boldsymbol{\delta}$ will be integrated into test-time examples to improve the prediction ability of $\boldsymbol{\theta}$.

**Adversarial visual prompting.** Inspired by the usefulness of VP to improve model generalization [Tsai et al., 2020, Bahng et al., 2022], we ask:

> **(AVP problem)** Can VP (1) be extended to robustify $\boldsymbol{\theta}$ against adversarial attacks?

At the first glance, the AVP problem seems trivial only if we specify the performance loss $\ell$ as the adversarial training (AT) loss [Madry et al., 2017, Zhang et al., 2019b]:

$$\ell_{\mathrm{adv}}(\mathbf{x} + \boldsymbol{\delta}; y, \boldsymbol{\theta}) = \underset{\mathbf{x}': \|\mathbf{x}' - \mathbf{x}\|_\infty \leq \epsilon}{\text{maximize}} \ell(\mathbf{x}' + \boldsymbol{\delta}; y, \boldsymbol{\theta}), \tag{2}$$

where $\mathbf{x}'$ denotes the adversarial input that lies in the $\ell_\infty$-norm ball centered at $\mathbf{x}$ with radius $\epsilon > 0$.

Recall from (1) that the conventional VP design requests $\boldsymbol{\delta}$ to be universal across training data. Thus, we term *universal AVP* (**U-AVP**) the following problem by integrating (1) with (2):

$$\underset{\boldsymbol{\delta}: \boldsymbol{\delta} \in \mathcal{C}}{\text{minimize}} \quad \lambda \mathbb{E}_{(\mathbf{x}, y) \in \mathcal{D}_{\mathrm{tr}}}[\ell(\mathbf{x} + \boldsymbol{\delta}; y, \boldsymbol{\theta})] + \mathbb{E}_{(\mathbf{x}, y) \in \mathcal{D}_{\mathrm{tr}}}[\ell_{\mathrm{adv}}(\mathbf{x} + \boldsymbol{\delta}; y, \boldsymbol{\theta})] \tag{U-AVP}$$

where $\lambda > 0$ is a regularization parameter to strike a balance between generalization and adversarial robustness [Zhang et al., 2019b].

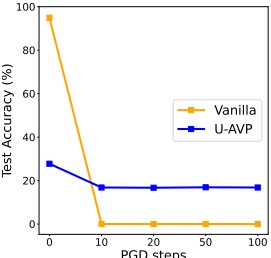

The problem (U-AVP) can be effectively solved using a standard min-max optimization method, which involves two alternating optimization routines: inner maximization and outer minimization. The former generates adversarial examples as AT, and the latter produces the visual prompt $\boldsymbol{\delta}$ like (1). At testing time, the effectiveness of $\boldsymbol{\delta}$ is measured from two aspects: (1) standard accuracy, *i.e.*, the accuracy of $\boldsymbol{\delta}$-integrated benign examples, and (2) robust accuracy, *i.e.*, the accuracy of $\boldsymbol{\delta}$-integrated adversarial examples (against the victim model $\boldsymbol{\theta}$). Despite the succinctness of (U-AVP), Fig. 1 shows its *ineffectiveness* to defend against adversarial attacks. Compared to the vanilla VP (1), it also suffers a significant standard accuracy drop (over $50\%$ in Fig. 1 corresponding to 0 PGD attack steps) and robust accuracy is only enhanced by a small margin (around $18\%$ against PGD attacks). The negative results in Fig. 1 are not quite surprising since a data-agnostic input prompt $\boldsymbol{\delta}$ has limited learning capacity to enable adversarial defense. Thus, it is non-trivial to tackle the problem of AVP.

Fig. 1: Example of designing U-AVP for adversarial defense on (CIFAR-10, ResNet18), measured by robust accuracy against PGD attacks [Madry et al., 2017] of different steps. The robust accuracy of 0 steps is the standard accuracy.

# 3 Class-wise Adversarial Visual Prompting

In this section, we will develop a new VP approach, termed Class-wise AVP (**C-AVP**), which improves (U-AVP) in adversarial robustness. Different from U-AVP, C-AVP expands the designing space of VP by associating each image class with an adversarial visual prompt and taking the couplings of these class-wise visual prompts into account for robustness enhancement.

**No free lunch for class-wise visual prompts.** A direct extension of (U-AVP) is to introduce multiple adversarial visual prompts, each of which corresponds to one class in the training set $\mathcal{D}_{\mathrm{tr}}$. If we split $\mathcal{D}_{\mathrm{tr}}$ into class-wise training sets $\{\mathcal{D}_{\mathrm{tr}}^{(i)}\}_{i=1}^{N}$ (for $N$ classes) and introduce class-wise visual prompts $\{\boldsymbol{\delta}^{(i)}\}$, then the direct C-AVP extension from (U-AVP) becomes

$$\underset{\{\boldsymbol{\delta}^{(i)}\in\mathcal{C}\}_{i\in[N]}}{\text{minimize}} \quad \frac{1}{N}\sum_{i=1}^{N}\left\{\lambda\mathbb{E}_{(\mathbf{x},y)\in\mathcal{D}_{\mathrm{tr}}^{(i)}}[\ell(\mathbf{x}+\boldsymbol{\delta}^{(i)};y,\boldsymbol{\theta})]+\mathbb{E}_{(\mathbf{x},y)\in\mathcal{D}_{\mathrm{tr}}^{(i)}}[\ell_{\mathrm{adv}}(\mathbf{x}+\boldsymbol{\delta}^{(i)};y,\boldsymbol{\theta})]\right\} \quad \text{(C-AVP-v0)}$$

where $[N]$ denotes the set of class labels $\{1, 2, \ldots, N\}$. It is worth noting that C-AVP-v0 is *decomposed* over class labels. That is, solving the above problem is equivalent to solving a sequence of sub-problems: For each class $i$,

$$\underset{\boldsymbol{\delta}^{(i)}\in\mathcal{C}}{\text{minimize}} \quad \lambda\mathbb{E}_{(\mathbf{x},y)\in\mathcal{D}_{\mathrm{tr}}^{(i)}}[\ell(\mathbf{x}+\boldsymbol{\delta}^{(i)};y,\boldsymbol{\theta})]+\mathbb{E}_{(\mathbf{x},y)\in\mathcal{D}_{\mathrm{tr}}^{(i)}}[\ell_{\mathrm{adv}}(\mathbf{x}+\boldsymbol{\delta}^{(i)};y,\boldsymbol{\theta})] \quad (3)$$

Although the class-wise separability facilitates numerical optimization, it introduces two challenges **(C1)**-**(C2)** when applying class-wise visual prompts to defend adversarial attacks.

• **(C1)** *Test-time prompt selection*: After acquiring the visual prompts $\{\boldsymbol{\delta}^{(i)}\}$ from (C-AVP-v0), it remains unclear how a class-wise prompt should be selected for application to a test-time example $\mathbf{x}_{\mathrm{test}}$. An intuitive way is to use the inference pipeline of $\boldsymbol{\theta}$ by aligning its top-1 prediction with the prompt selection. That is, the selected prompt $\boldsymbol{\delta}$ and the predicted class $i^*$ are determined by

$$\boldsymbol{\delta} = \boldsymbol{\delta}^*, \ i^* = \underset{i\in[N]}{\arg\max} f_i(\mathbf{x}_{\mathrm{test}}+\boldsymbol{\delta}^{(i)};\boldsymbol{\theta}), \quad (4)$$

where $f_i(\mathbf{x};\boldsymbol{\theta})$ denotes the $i$th-class prediction confidence of using $\boldsymbol{\theta}$ at $\mathbf{x}$. However, the seemingly correct rule (4) leads to a large prompt selection error (thus poor prediction accuracy) due to **(C2)**.

• **(C2)** *Backdoor effect of class mis-matched prompts*: Given $\boldsymbol{\delta}^{(i)}$ from (3), if the test-time example $\mathbf{x}_{\mathrm{test}}$ is drawn from class $i$, the visual prompt $\boldsymbol{\delta}^{(i)}$ then helps prediction. However, if $\mathbf{x}_{\mathrm{test}}$ is *not* originated from class $i$, then $\boldsymbol{\delta}^{(i)}$ could serve as a backdoor attack trigger [Gu et al., 2017] with the targeted backdoor label $i$ for the 'prompted input' $\mathbf{x}_{\mathrm{test}}+\boldsymbol{\delta}^{(i)}$. Since the backdoor attack is also input-agnostic, the class-discriminative ability of $\mathbf{x}_{\mathrm{test}}+\boldsymbol{\delta}^{(i)}$ enabled by $\boldsymbol{\delta}^{(i)}$ could result in incorrect prediction towards the target class $i$ for $\mathbf{x}_{\mathrm{test}}$. Our empirical experiments justified the above: Nearly all testing samples will be (mis)classified as the prompt's class regardless of their true labels.

**Joint prompts optimization for C-AVP.** The failure of C-AVP-v0 inspires us to re-think the value of class-wise separability in (3). As illustrated in challenges **(C1)**-**(C2)**, the compatibility with the test-time prompt selection rule and the interrelationship between class-wise visual prompts should be taken into account. To this end, we develop a series of new AVP principles below. Fig. 2 provides a schematic overview of C-AVP and its comparison with U-AVP and the original predictor without VP.

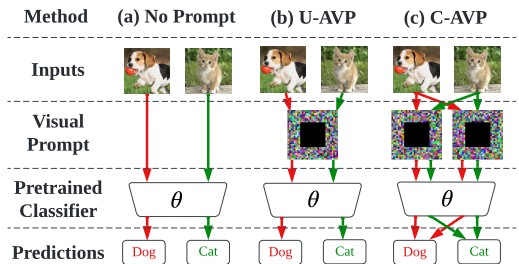

Fig. 2: Overview of C-AVP over two classes (red and green) vs. U-AVP and the prompt-free learning pipeline.

First, to bake the prompt selection rule (4) into C-AVP, we enforce the prompt design along the correct selection path, *i.e.*, under the condition that $f_y(\mathbf{x}+\boldsymbol{\delta}^{(y)};\boldsymbol{\theta}) > \max_{k:k\neq y} f_k(\mathbf{x}+\boldsymbol{\delta}^{(k)};\boldsymbol{\theta})$ for $(\mathbf{x},y)\in\mathcal{D}^{(y)}$. The above can be cast as a CW-type loss [Carlini and Wagner, 2017]:

$$\ell_{\mathrm{C-AVP},1}(\{\boldsymbol{\delta}^{(i)}\};\mathcal{D}_{\mathrm{tr}},\boldsymbol{\theta}) = \mathbb{E}_{(\mathbf{x},y)\in\mathcal{D}_{\mathrm{tr}}}\max\{\max_{k\neq y}f_k(\mathbf{x}+\boldsymbol{\delta}^{(k)};\boldsymbol{\theta})-f_y(\mathbf{x}+\boldsymbol{\delta}^{(y)};\boldsymbol{\theta}),-\tau\}, \quad (5)$$

where $\tau > 0$ is a confidence threshold. The rationale behind (5) is that given a data sample $(\mathbf{x}, y)$, the minimum value of $\ell_{\text{C-AVP},1}$ is achieved at $-\tau$, indicating the desired condition with the confidence level $\tau$. Compared with (C-AVP-v0), another key characteristic of $\ell_{\text{C-AVP},1}$ is its non-splitting over class-wise prompts $\{\boldsymbol{\delta}^{(i)}\}$, which benefits the joint optimization of these prompts.

Second, to mitigate the backdoor effect of class mis-matched prompts, we propose additional two losses, noted by $\ell_{\text{C-AVP},2}$ and $\ell_{\text{C-AVP},3}$, to penalize the data-prompt mismatches. Specifically, $\ell_{\text{C-AVP},2}$ penalizes the backdoor-alike targeted prediction accuracy of a class-wise visual prompt when applied to mis-matched training data. For the prompt $\boldsymbol{\delta}^{(i)}$, this leads to

$$\ell_{\text{C-AVP},2}(\{\boldsymbol{\delta}^{(i)}\};\mathcal{D}_{\text{tr}},\boldsymbol{\theta}) = \frac{1}{N}\sum_{i=1}^{N}\mathbb{E}_{(\mathbf{x},y)\in\mathcal{D}_{\text{tr}}^{(-i)}}\max\{f_i(\mathbf{x}+\boldsymbol{\delta}^{(i)};\boldsymbol{\theta}) - f_y(\mathbf{x}+\boldsymbol{\delta}^{(i)};\boldsymbol{\theta}), -\tau\}, \quad (6)$$

where $\mathcal{D}_{\text{tr}}^{(-i)}$ denotes the training data set by excluding $\mathcal{D}_{\text{tr}}^{(i)}$. The rationale behind (6) is that the class $i$-associated prompt $\boldsymbol{\delta}^{(i)}$ should *not* behave as a backdoor trigger to non-$i$ classes' data. Likewise, if the prompt is applied to the correct data class, then the prediction confidence of this matched case should surpass that of a mis-matched case. This leads to

$$\ell_{\text{C-AVP},3}(\{\boldsymbol{\delta}^{(i)}\};\mathcal{D}_{\text{tr}},\boldsymbol{\theta}) = \mathbb{E}_{(\mathbf{x},y)\in\mathcal{D}_{\text{tr}}}\max\{\max_{k\neq y} f_y(\mathbf{x}+\boldsymbol{\delta}^{(k)};\boldsymbol{\theta}) - f_y(\mathbf{x}+\boldsymbol{\delta}^{(y)};\boldsymbol{\theta}), -\tau\}. \quad (7)$$

Let $\ell_{\text{C-AVP},0}(\{\boldsymbol{\delta}^{(i)}\};\mathcal{D}_{\text{tr}},\boldsymbol{\theta})$ denote the objective function of (C-AVP-v0). Integrated with $\ell_{\text{C-AVP},q}(\{\boldsymbol{\delta}^{(i)}\};\mathcal{D}_{\text{tr}},\boldsymbol{\theta})$ for $q \in \{1,2,3\}$, the desired class-wise AVP design is cast as

$$\minimize_{\{\boldsymbol{\delta}^{(i)}\in\mathcal{C}\}_{i\in[N]}} \quad \ell_{\text{C-AVP},0}(\{\boldsymbol{\delta}^{(i)}\};\mathcal{D}_{\text{tr}},\boldsymbol{\theta}) + \gamma\sum_{q=1}^{3}\ell_{\text{C-AVP},q}(\{\boldsymbol{\delta}^{(i)}\};\mathcal{D}_{\text{tr}},\boldsymbol{\theta}), \quad \text{(C-AVP)}$$

where $\gamma > 0$ is a regularization parameter to control our emphasis on the class-wise prompting penalties. It is worth mentioning that since $\ell_{\text{C-AVP},q}$ (for $q > 0$) is a hinge loss with hard threshold $\tau$, its optimization could automatically stop if a prompting regulation is satisfied. To solve (C-AVP), we will use the min-max optimizer similar to the approach used for solving (C-AVP-v0).

## 4 Experiments

**Experiment setup.** We conduct experiments on CIFAR-10 with a pretrained ResNet18 of testing accuracy of 94.92% on standard test dataset. We use PGD-10 (*i.e.*, PGD attack with 10 steps [Madry et al., 2017]) to generate adversarial examples with $\epsilon = 8/255$ during visual prompts training, and with a cosine learning rate scheduler starting at $0.1$. Throughout experiments, we choose $\lambda = 1$ in (U-AVP), and $\tau = 0.1$ and $\gamma = 3$ in (C-AVP). The width of visual prompt is set to 8 (see Fig. 3). To evaluate test-time adversarial robustness, we generate PGD attacks of different steps under $\epsilon = 8/255$.

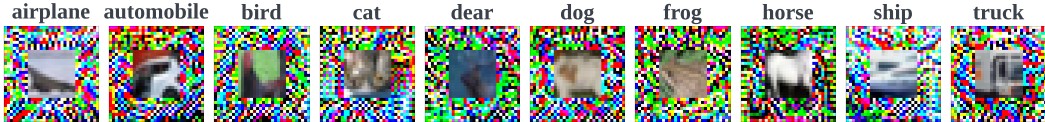

Fig. 3: C-AVP visualization. One image is chosen from each CIFAR-10 class with the corresponding C-AVP.

**C-AVP outperforms conventional VP.** Tab. 1 demonstrates the effectiveness of proposed C-AVP approach vs. U-AVP (the direction extension of VP to adversarial defense) and the C-AVP-v0 method in the task of robustify a normally-trained ResNet18 on CIFAR-10. For comparison, we also report the standard accuracy of the pre-trained model and the vanilla VP solution given by (1). As we can see, C-AVP outperforms U-AVP and C-AVP-v0 in both standard accuracy and robust accuracy (evaluated using PGD attacks with different step sizes). We also observe that compared to the pre-trained model and the vanilla

Table 1: VP performance comparison in terms of standard (std) accuracy (acc) and robust accuracy against PGD attacks with $\epsilon = 8/255$ and multiple PGD steps on (CIFAR-10, ResNet18).

| Evaluation metrics (%) | Std acc | Robust acc vs PGD w/ step # | | | |
|---|---|---|---|---|---|
| | | 10 | 20 | 50 | 100 |
| Pre-trained | **94.92** | 0 | 0 | 0 | 0 |
| Vanilla VP | 94.48 | 0 | 0 | 0 | 0 |
| U-AVP | 27.75 | 16.9 | 16.81 | 16.81 | 16.7 |
| C-AVP-v0 | 19.69 | 13.91 | 13.63 | 13.6 | 13.58 |
| C-AVP (ours) | 57.57 | **34.75** | **34.62** | **34.51** | **33.63** |

VP, the robustness-induced VP variants bring in an evident standard accuracy drop as the cost of robustness enhancement. This leaves a future research direction to optimize the accuracy-robustness trade-off of visual prompts.

**Prompting regularization effect in** (C-AVP). Tab. 2 shows different settings of prompting regularizations used in C-AVP, where 'S$i$' represents a certain loss configuration. As we can see, the use of $\ell_{\text{C-AVP},2}$ contributes most to improving the performance of learned visual prompts (see S3). This is not surprising, since we design $\ell_{\text{C-AVP},2}$ for mitigating the backdoor effect of class-wise prompts, which is the main source of

Table 2: Sensitivity analysis of prompting regularizations in C-AVP on (CIFAR-10, ResNet18).

| Setting | $\ell_{\text{C-AVP},1}$ | $\ell_{\text{C-AVP},2}$ | $\ell_{\text{C-AVP},3}$ | Std Acc (%) | PGD-10 Acc (%) |
|---------|------|------|------|-------|-------|
| S1 | ✗ | ✗ | ✗ | 19.69 | 13.91 |
| S2 | ✔ | ✗ | ✗ | 22.72 | 13.01 |
| S3 | ✗ | ✔ | ✗ | 40.01 | 25.40 |
| S4 | ✗ | ✗ | ✔ | 17.44 | 11.78 |
| S5 | ✔ | ✔ | ✗ | 57.03 | 32.39 |
| S6 | ✔ | ✗ | ✔ | 26.02 | 15.80 |
| S7 | ✔ | ✔ | ✔ | **57.57** | **34.75** |

prompting selection error. We also note that $\ell_{\text{C-AVP},1}$ is the second most important regularization, as evidenced by the comparable performance of S3 vs. S5. This is because such a regularization is accompanied with the prompt selection rule (4). If training cost is taken into consideration, Tab. 2 also indicates that the combination of $\ell_{\text{C-AVP},1}$ and $\ell_{\text{C-AVP},2}$ is a possible computationally lighter alternative to (C-AVP).

**Class-wise prediction error analysis.** Fig. 4 shows a comparison of the classification confusion matrix over benign test dataset. Here the row index signifies the test data per class, and the column index refers to the selected prompt for prediction when using C-AVP-v0 or C-AVP. As we can see, our proposal outperforms C-AVP-v0 since the former's higher

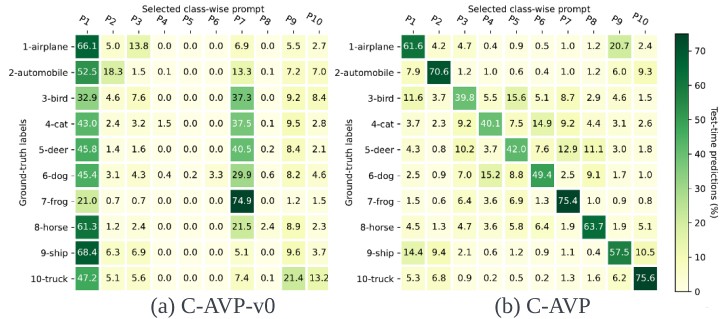

(a) C-AVP-v0      (b) C-AVP

Fig. 4: The prediction error analysis of C-AVP vs. C-AVP-v0 on (CIFAR10, ResNet18). Each row corresponds to testing samples from one class, and each column corresponds to the prompt ('P') selection across 10 image classes.

main diagonal entries indicate better prompt selection accuracy (and thus prediction accuracy) than the latter. In Fig. 4(b), we also observe that the incorrect class-wise predictions (*i.e.*, the off-diagonal entries) often appear for similar classes such as (class 1-airplane, P1) and (class 9-ship, P9).

**Comparisons with other test-time defenses.** In Tab. 3, we compare our proposed C-AVP with three test-time defense methods, selected from [Croce et al., 2022]. Note that all methods are applied to robustifying a fixed, normally pre-trained ResNet18. Following [Croce et al., 2022], we divide the considered defenses into different categories, relying on their defense principles (*i.e.*, IP or MA) as well as their needed test-time operations (*i.e.*, IA, AN, and R). As we can see, our method C-AVP falls into the IP category but requires no involved test-time

Table 3: Comparison of C-AVP with other SOTA test-time defenses. Per the benchmark in [Croce et al., 2022], the involved test-time operations in these defenses include: IP (input purification), MA (model adaption), IA (iterative algorithm), AN (auxiliary network), and R (randomness). And inference time (IT), standard accuracy (SA), and robust accuracy (RA) against PGD-10 are used as performance metrics.

| Method | IP | MA | IA | AN | R | IT | SA (%) | RA (%) |
|--------|----|----|----|----|---|-----|--------|--------|
| Shi et al. [2021] | ✔ | ✗ | ✔ | ✗ | ✗ | 518 × | 85.9% | 0.4% |
| Yoon et al. [2021] | ✔ | ✗ | ✔ | ✔ | ✔ | 176 × | 91.1% | 40.3% |
| Chen et al. [2021] | ✗ | ✔ | ✔ | ✔ | ✗ | 59 × | 56.1% | 50.6% |
| C-AVP | ✔ | ✗ | ✗ | ✗ | ✗ | **1.4 ×** | 57.6% | 34.3% |

operations. This leads to the least inference overhead. Although there exists a performance gap with the test-time defense baselines, we hope that our work could pave a way to study the pros and cons of visual prompting in adversarial robustness.

## 5 Conclusion

In this work, we develop a novel VP method, *i.e.*, C-AVP, to improve adversarial robustness of a fixed model at testing time. Compared to existing VP methods, this is the first work to peer into how VP could be in adversarial defense. We show that the direct integration of VP into robust learning does *not* offer an effective adversarial defense at testing time for the fixed model. To address this problem, we propose C-AVP to create ensemble visual prompts and jointly optimize their interrelations for robustness enhancement. We empirically show that our proposal significantly reduces the inference overhead compared to classical adversarial defenses which typically call for computationally-intensive test-time defense operations.

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
