# OpenReview forum: "Visual Prompting for Adversarial Robustness"
_NeurIPS.cc/2022/Workshop/TSRML — TSRML2022_

### Official Review · Reviewer_XNDP · 2022-10-12
**Visual prompts - An interesting novel method for improving neural networks' adversarial robustness**

**Overall Rating:** 7

**Summary:**

The paper considers Visual Prompts (VP) for increasing neural network's adversarial robustness. The authors argue that universal VPs are not well suited for increasing the adversarial robustness of neural networks. Instead the authors use Class-wise Visual Prompts (CVP) and a new optimisation method for obtaining these CVPs. The paper presents empirical results to show that (i) vanilla VPs are not well suited for improving the adversarial robustness, (ii) naive approaches for producing CVPs result in sub-optimal results for adversarial robustness, and (iii) the method proposed here produces visual prompts that significantly improve the adversarial robustness.

**Strengths:**

The idea of using visual prompts to improve the adversarial robustness is interesting and, to the best of my knowledge, novel. The topic is timely and a good fit for the workshop.

The paper is well written and easy to follow, moreover it provides sufficient amount of details for reproducabiltiy.

Empirical results show significant improvements in adversarial accuracy compared to non-defended networks and networks defended with naive visual prompts. As the method presented here differs significantly from most other state-off-the-art defense methods, a direct comparission is difficult. However, the results presented in the paper indicate a good performance compared to other defense methods.

**Weaknesses:**

1) It is not clear to me whether the adversarial attacks are performed on the original image (attackers are unaware of the defense method) or the adversarial attacks are performed on the image with the VPs (attackers are aware of the defense method). While both attack methods are of interest, the treat model under consideration should be clearly stated. Moreover, I would consider an evaluation w.r.t. both attack methods to strengthen the paper, even if performance is significantly reduced for the latter attack.

2) The experimental evaluation is somewhat limited. It would be interesting to see results for a range of visual-prompt widths. Moreover, it would be interesting to see whether this approach still works for e.g. imagenet, where the object of interest is less consistently focused at the center of the image than for Cifar-10.

Minor comments:

1) Deer is misspelt as "dear" in Figure 3.

2) Figure 1 could state the perturbation radii (epsilon).

3) The current formulation makes it a bit challenging to understand the idea behind the losses defined in equations 6 & 7, you might want to try to expand this discussion a bit.


**Overall Recommendation:**

A strong submission with clear and significant novel contributions. The strengths of the paper outweigh the weakness of a somewhat limited empirical evaluation.

**Review Confidence:**

4: The reviewer is confident but not absolutely certain that the evaluation is correct

---

### Official Review · Reviewer_KaZ3 · 2022-10-18
**An interesting approach for an important problem**

**Overall Rating:** 6

**Summary:**

The authors suggest the application of the existing visual prompting (VP) methodology to the case of adversarial robustness. They illustrate that straightforward adaptations result in poor performance and provide several improvements to make VP more suitable for the task. They achieve significant boost in inference latency compared to the existing work while showing decent performance in terms of clean and robust accuracy.

**Strengths:**


**Quality**

The paper is properly formatted and structured. The method discussion and the experiments are sound. The introduction of additional loss terms is well justified and supported by ablation studies (Table 2).

**Clarity**

The work is clear and easy to follow. It contains a lot of illustrations that support the discussion.

**Originality**

Up to the reviewer's knowledge, this is the first attempt to apply visual prompting to the problem of adversarial robustness

**Significance**

Test-time defenses are an important direction given the computational overhead of adversarial training and similar techniques. Therefore, research in this direction is of great value.

**Weaknesses:**


Q1. Evaluation with the standard PGD [1] in Tables 1-3 is not very informative. More sophisticated evaluation techniques such as AutoAttack [2] (or at least A-PGD) would be preferable.

[1] Aleksander Madry, Aleksandar Makelov, Ludwig Schmidt, Dimitris Tsipras, and Adrian Vladu. "Towards deep learning models resistant to adversarial attacks". In ICLR 2018.

[2] Francesco Croce and Matthias Hein. "Reliable evaluation of adversarial robustness with an ensemble of diverse parameter-free attacks". In ICML 2020

Q2. Line 199: "The width of visual prompt is set to 8". How are the shape and size for the visual prompt chosen? Is it a standard choice accepted in previous work? Why is the center of the image not affected? Does it assume that the center contains the information for properly predicting the class?

Q3. Given that U-AVP was demonstrated to be inefficient (Figure 1), transitioning to a set of VPs instead of a single VP seems like a logical step. What is the motivation for connecting it with the set of classes? One problem could be that if we transition from CIFAR-10 (10 classes) to ImageNet-1K (1000 classes), then the computational overhead at inference time may become too big. Are there other ways to define a set of VPs that would allow to compromise between perfomance and computational efficiency?

**Overall Recommendation:**

The paper suggests a novel application of visual prompting to an important problem of adversarial robustness and achieves impressive results in terms of inference time. The method description and the experiments are sound. Therefore, I recommend accepting this paper.

**Review Confidence:**

4: The reviewer is confident but not absolutely certain that the evaluation is correct

---

### Official Review · Reviewer_wpHz · 2022-10-22
**New approach in defending against adversarial examples using visual prompts**

**Overall Rating:** 7

**Summary:**

This paper investigates suitability of visual prompts as a defense against adversarial examples. It first highlights the limitation of universal prompts as a defense and later proposes class-wise adversarial visual prompts (C-AVP) as a solution.


**Strengths:**

- Paper is very well written and easy to understand.
- Code is publicly available.
- First of its kind work on using AVP as an defense

**Weaknesses:**

- The main concern is the dependence of proposed visual prompting approach on number of classes. Wouldn’t a high number of classes, e.g., 1000 for imagenet [1], lead to much higher computational cost in optimizing for visual prompt.
- Similarly the evaluation time for the prompting approach increases linearly with number of classes. With 1000 classes for ImageNet, it would incur 1000x evaluation cost too.

1. Deng, Jia, Wei Dong, Richard Socher, Li-Jia Li, Kai Li, and Li Fei-Fei. "Imagenet: A large-scale hierarchical image database." In 2009 IEEE conference on computer vision and pattern recognition, pp. 248-255. Ieee, 2009.


**Overall Recommendation:**

Though the proposed method heavily underperforms, I recommend acceptance since it opens up a new line of investigation in use of visual prompts for adversarial robustness.


**Review Confidence:**

4: The reviewer is confident but not absolutely certain that the evaluation is correct

---

### Decision · Program_Chairs · 2022-10-23

Accept